# Neutralization of SARS-CoV-2 spike pseudotyped virus by recombinant ACE2-Ig

Changhai Lei[1,2], Kewen Qian[1,2], Tian Li[1,2], Sheng Zhang[3], Wenyan Fu[4], Min Ding[5] & Shi Hu[1,2 ✉]

Severe acute respiratory syndrome coronavirus 2 (SARS-CoV-2) emerged in Wuhan, China, at the end of 2019, and there are currently no specific antiviral treatments or vaccines available. SARS-CoV-2 has been shown to use the same cell entry receptor as SARS-CoV, angiotensin-converting enzyme 2 (ACE2). In this report, we generate a recombinant protein by connecting the extracellular domain of human ACE2 to the Fc region of the human immunoglobulin IgG1. A fusion protein containing an ACE2 mutant with low catalytic activity is also used in this study. The fusion proteins are then characterized. Both fusion proteins have a high binding affinity for the receptor-binding domains of SARS-CoV and SARS-CoV-2 and exhibit desirable pharmacological properties in mice. Moreover, the fusion proteins neutralize virus pseudotyped with SARS-CoV or SARS-CoV-2 spike proteins in vitro. As these fusion proteins exhibit cross-reactivity against coronaviruses, they have potential applications in the diagnosis, prophylaxis, and treatment of SARS-CoV-2.

[1] Department of Biophysics, College of Basic Medical Sciences, Second Military Medical University, Shanghai 200433, China. [2] Team SMMU-China of the International Genetically Engineered Machine (iGEM) competition, Department of Biophysics, Second Military Medical University, Shanghai 200433, China. [3] Department of Critical Care Medicine, Ruijin Hospital, Shanghai Jiao Tong University School of Medicine, Shanghai 200011, China. [4] Department of Assisted Reproduction, Shanghai Ninth People's Hospital, Shanghai Jiao Tong University School of Medicine, Shanghai 200011, China. [5] Pharchoice Therapeutics, Inc, Shanghai 201406, China. ✉email: hus@smmu.edu.cn

An outbreak of coronavirus disease 2019 (COVID-19) caused by the 2019 novel coronavirus (severe acute respiratory syndrome coronavirus 2, SARS-CoV-2) that emerged in Wuhan, China, at the end of 2019, has widely spread worldwide. The full spectrum of COVID-19 ranges from mild, self-limiting respiratory tract illness to severe progressive pneumonia, multiorgan failure, and death[1]. However, there are currently no specific antiviral treatments or vaccines for SARS-CoV-2. The spike (S) proteins of coronaviruses, including the coronavirus that causes severe acute respiratory syndrome (SARS) and SARS-CoV-2, associate with cellular receptors to mediate the infection of their target cells. Angiotensin-converting enzyme 2 (ACE2), a carboxypeptidase that potently degrades angiotensin II to angiotensin 1–7, has been identified as a functional receptor for SARS-CoV[2] and a potent receptor for SARS-CoV-2[3,4]. This metallopeptidase also plays a key role in the renin–angiotensin system (RAS)[5].

As ACE2, a key player in hormone cascades, plays a central role in the homeostatic control of cardiorenal actions, researchers carried out pharmacodynamics studies of recombinant ACE2 (rACE2) and determined its ability to protect against severe acute lung injury and acute Ang II-induced hypertension[6–8]. In mouse models, the administration of rACE2 was also shown to inhibit myocardial remodeling and attenuate Ang II-induced cardiac hypertrophy and cardiac dysfunction[9], as well as renal oxidative stress, inflammation, and fibrosis[10,11]. Moreover, recombinant ACE2 protein was shown to have therapeutic potential for the SARS-CoV[12]. However, in both human and mice[13,14], rACE2 exhibits a fast clearance rate, with a half-life of only hours reported by pharmacokinetic studies. Recently, a fusion protein consisting of murine rACE2 with a Fc fragment (rACE2-Fc) showed the ability to protect organs in both acute and chronic models of angiotensin II-dependent hypertension in mice[15]; interestingly, the fusion protein also had long-lasting effects.

Here, we show that recombinant protein of the extracellular domain of human ACE2 fused with the Fc region of the human immunoglobulin IgG1 (termed as ACE2-Ig) shows high-affinity binding to the receptor-binding domain (RBD) of SARS-CoV and SARS-COV-2 and exerted desired pharmacological properties. Moreover, ACE2-Ig shows potent cross-reactivity against both SARS-CoV and SARS-CoV2 in vitro. Thus, our data support further investigation of ACE2-Ig for diagnosis, prophylaxis, and treatment of SARS-COV-2.

## Results

**Generation of ACE2-Ig.** Based on the receptor function of ACE2 for coronaviruses, we hypothesized that the ACE2 fusion protein has the potential to neutralize coronaviruses, especially SARS-CoV-2. To further evaluate the therapeutic potential of ACE2, a fusion protein (ACE2-Ig) consisting of the extracellular domain of human ACE2 linked to the Fc domain of human IgG1 was designed and generated (Fig. 1; Supplementary Table 1). An ACE2 variant (HH/NN) in which two active-site histidine residues (residues 374 and 378) had been altered to asparagine residues to reduce the catalytic activity was also used in our study. The fusion protein containing the ACE2 variant was termed

mACE2-Ig. Expression and purification were carried out as described in our previous reports[16,17]. The affinities of the fusion proteins for the SARS-CoV RBD and SARS-CoV-2 RBD were determined with BIAcore-binding assays (Supplementary Fig. 1), and both fusion proteins showed a high affinity for the RBDs. Moreover, the fusion proteins had denaturation temperatures similar to those of the Fc fusion protein TIGIT-Ig, as reported in our previous study[17], suggesting IgG-like stability. The lowest concentration (< 2%) of high- and low-molecular-weight products was observed after 1 week of storage at 40 °C at a 1 mg ml$^{-1}$ concentration. The pharmacokinetic (PK) parameters of the fusion proteins were then tested in our study. Mice were separately treated with a single intravenous dose of the fusion proteins, and the serum concentrations of the fusion proteins were determined by ELISA. The results showed that the main PK parameters of ACE2-Ig, mACE2-Ig, and TIGIT-Ig in mice were very similar and demonstrated the high stability of the fusion proteins. The experimental data are summarized in Supplementary Table 2.

**Neutralization effect of ACE2-Ig.** After we identified that the ACE2 fusion proteins bind the RBDs with a high affinity, we next sought to test the inhibitory activities of the ACE2 fusion proteins against SARS-CoV-2 and compare them with those against SARS-CoV by using viruses pseudotyped with the S glycoproteins of SARS-CoV and SARS-CoV-2. Our data show that both SARS-CoV and SARS-CoV-2 were potently neutralized by ACE2-Ig and mACE2-Ig. The IC$_{50}$ values of ACE2-Ig for SARS-CoV and SARS-CoV-2 neutralization were 0.8 and 0.1 µg ml$^{-1}$, respectively. The IC$_{50}$ values of mACE2-Ig for the neutralization of the two viruses were 0.9 and 0.08 µg ml$^{-1}$, respectively (Fig. 2a). Moreover, in A549 cells, The IC$_{50}$ values of ACE2-Ig for SARS-CoV and SARS-CoV-2 neutralization were 1.4 and 0.1 µg ml$^{-1}$, respectively. The IC$_{50}$ values of mACE2-Ig for the neutralization of the two viruses were 0.6 and 0.03 µg ml$^{-1}$, respectively (Fig. 2b). No evidence of neutralization was observed for TIGIT-Ig (Supplementary Fig. 2). We next used a cell fusion assay to further characterize the in vitro neutralizing effects of the fusion proteins (Fig. 3). ACE2-Ig potently inhibited SARS-CoV S protein-mediated fusion with an IC$_{50}$ value of 0.85 µg ml$^{-1}$ and SARS-CoV-2 S protein-mediated fusion with an IC$_{50}$ value of 0.65 µg ml$^{-1}$. Under the same experimental conditions, another fusion protein, mACE2-Ig, exhibited IC$_{50}$ values of 0.76 µg ml$^{-1}$ and 0.48 µg ml$^{-1}$ for SARS-CoV and SARS-CoV-2, respectively. The control, TIGIT-Fc, did not show any inhibitory effect in this assay. These data suggest that both ACE2-Ig and mACE2-Ig exhibit potent inhibitory activity against SARS-CoV and SARS-CoV-2. In both assays, we observed that ACE2-Ig inhibited SARS-CoV-2 S protein-driven entry with higher efficiency than SARS-CoV S protein-mediated entry. We therefore sought to determine the binding kinetics for ACE2-Ig and nCoV S proteins. The affinity of SARS-CoV-2 S protein ACE2-Ig bound to ACE2-Ig is ~10 nM, which is ~16-fold higher than that of SARS-CoV S protein bound to ACE2-Ig (Fig. 3b), which is consistent with a very recent study[4]. The higher affinity of SARS-COV-2 S for ACE2-Ig may contribute to the apparent higher efficiency of neutralization.

## Discussion

ACE2 has already been identified as an important drug target for the treatment of cardiovascular and kidney diseases. Although ACE2 is a key functional receptor for coronavirus infection, practical use of the rACE2 protein may be impeded by the short half-life of rACE2. Recombinant Fc fusion, which can extend the plasma residence time of soluble proteins, improve in vivo

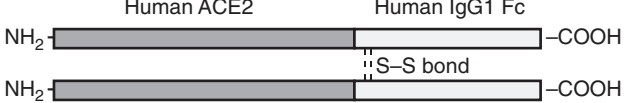

**Fig. 1 Schematic of ACE2-Ig.** The recombinant ACE2-Ig fusion protein exists as a homodimer with each subunit consisting of a ACE2 peptidase domain linked to an Fc domain of human IgG.

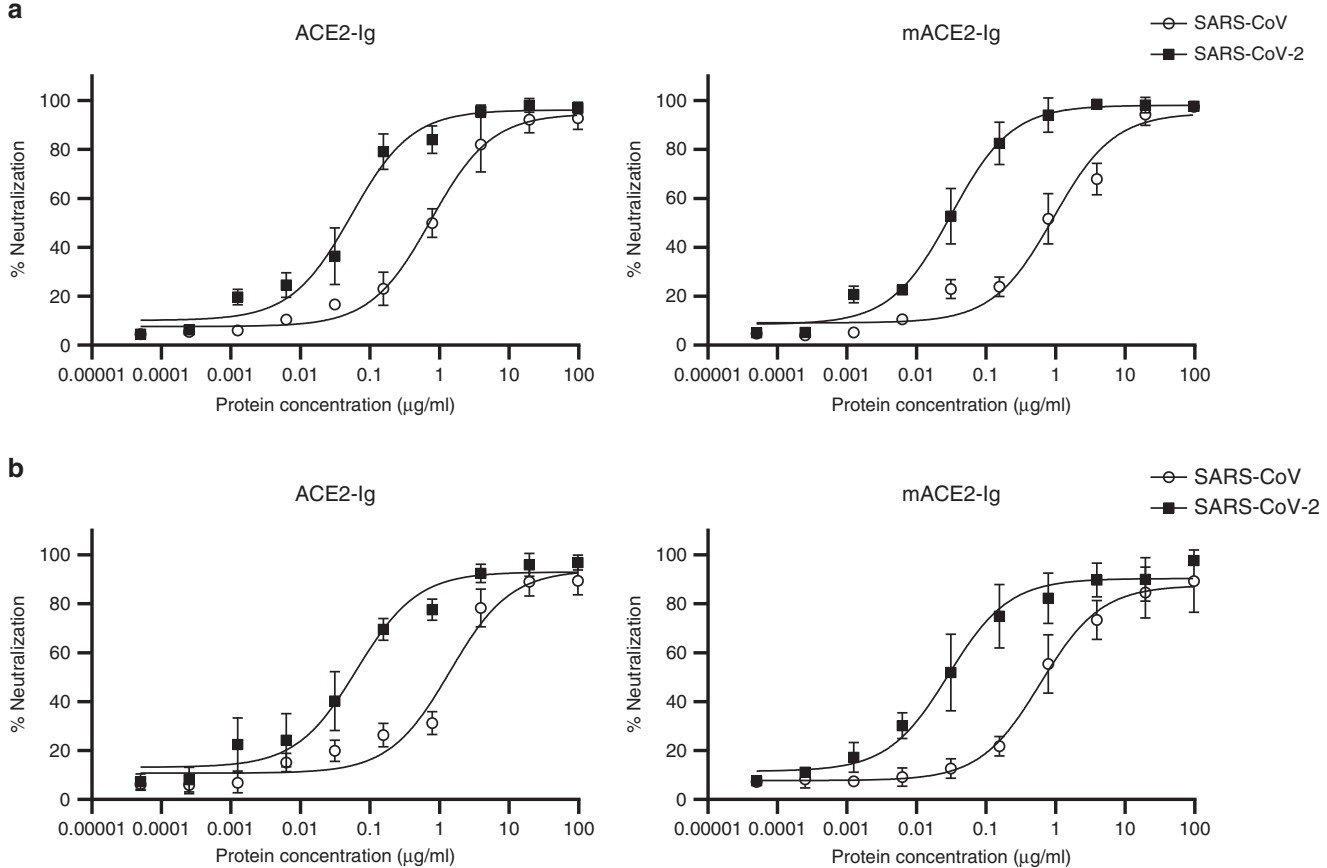

**Fig. 2 ACE2-Ig potently neutralizes viruses pseudotyped with the S glycoproteins.** HIVs pseudotyped with the S glycoproteins from CoVs were incubated with different fusion proteins for 1 h before infection. Luciferase activities in target 293 T cells (**a**) and A549 cells (**b**) were measured, and the percent neutralization was calculated. Data are the means ± s.d. of four independent biological replicates. Source data (**a**, **b**) are provided as a Source Data file.

efficacy, and gain immunoreactive functions, has been widely used in modern biopharmaceuticals. For example, long-acting forms of both the recombinant coagulation factors rFVIII-Fc and rFIX-Fc were recently approved for the clinical treatment of hemophilia A and B, requiring less frequent infusions[18,19]. In addition, a TNF receptor fusion protein has been used in the clinic for decades[20]. Unlike coagulation factors, which are blood-resident enzymes, full-length endogenous ACE2 is a transmembrane protein anchored to the cell surface, and ACE2 activity is, in fact, present at very low levels in the systemic circulation[21–23]. Therefore, one safety concern for the systemic administration of ACE2 fusion proteins is that they may have systemic cardiovascular side effects. Interestingly, one recent report showed no evidence of side effects in mice treated with murine ACE2 fusion proteins[15]. Moreover, our preliminary studies showed that the neutralizing effect of ACE2 remained when two active-site histidine residues of ACE2 were modified to asparagine residues.

The rapid global spread of SARS-CoV2 highlights the urgent need for coronavirus vaccines and therapeutics. Despite the relatively high degree of structural homology between the SARS-CoV-2 RBD and the SARS-CoV RBD[24], most of the currently known human antibodies with potent neutralizing activity to the SARS-CoV, including S230, m396 and 80 R, show no cross-reactivity to SARS-CoV-2[24]. We provide evidence that neutralization of SARS-CoV-2 with ACE2-Ig can be achieved, but our in vitro efficacy model is based on the pseudovirus system. Although this assay is sensitive and quantitative in previous reports[25], it may not fully recapitulate the live virus. In summary,

our data reveal the therapeutic potential of ACE2-based strategies, which can be used alone or in combination, against SARS-CoV-2. Based on the molecular mechanisms of ACE2 blockade, these fusion proteins may have broad neutralizing activity against coronavirus. These ACE2 fusion proteins could also be used for diagnosis and as research reagents in the development of vaccines and inhibitors.

## Methods

**Generation of fusion proteins.** To generate the recombinant fusion plasmids, the DNA sequences of the extracellular domains (ECDs) of *ACE2* were ligated to the DNA sequence of the Fc segment of human IgG1. Mutations were generated by Integrated DNA Technologies. We used the pcDNA3.4 vector and FreeStyle 293 expression system (Invitrogen) to express the fusion proteins. The fusion proteins were further purified using protein A-Sepharose with the harvested cell culture supernatants. TIGIT-Ig[17] served as a control in our study. The concentrations and purity of the fusion proteins were determined by measuring the UV absorbance at a wavelength of 280 nm and by polyacrylamide gel electrophoresis, respectively.

**Affinity measurement.** Immobilized anti-human Fc polyclonal antibody (Jackson ImmunoResearch, 109-005-008) on a CM5 chip (~150 RU) was used to capture the fusion proteins and then injected CoV RBDs[26] (12.5 nM–200 nM). Blank flow cells were used for correction of the binding response. The surface plasmon resonance (SPR) method was used with a BIAcore 2000 to measure the monovalent binding affinities of the fusion proteins, and kinetic analysis was performed using a 1:1 L model that simultaneously fits ka and kb.

**Pharmacokinetics.** In vivo experiments were approved by the Institutional Animal Care and Use Committee (IACUC) of Second Military Medical University, and mice were housed in a specific pathogen-free barrier facility. We used BALB/c mice to determine the pharmacokinetic profiles of the fusion proteins. Eight-week-old

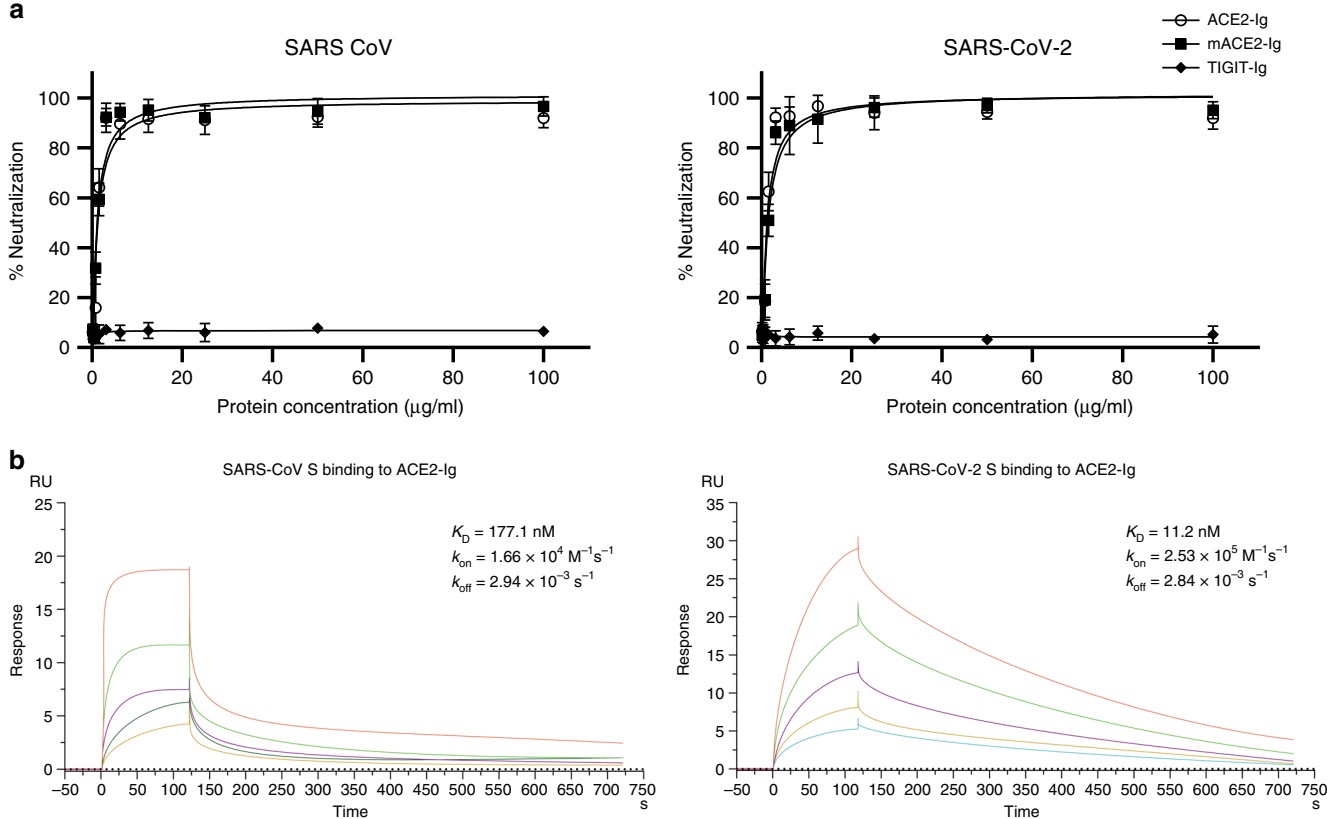

**Fig. 3 Inhibition of cell fusion by ACE2-Ig. a** Potent inhibition of cell fusion was mediated by the SARS-CoV (left) or SARS-CoV-2 (right) S glycoproteins. Cells expressing different S glycoproteins were incubated with the indicated fusion protein and mixed with ACE2-expressing cells. The activity of the reporter gene, β-gal, was measured as a correlate of fusion. The curves represent the best fit to the experimental data, and were used to calculate $IC_{50}$ values. **b** Kinetic analysis of SARS-CoV S protein and SARS-CoV-2 S protein binding to ACE2-Ig was performed by surface plasmon resonance (SPR). Data are the means ± s.d. of four (**a**) independent biological replicates. Results shown represent three (**b**) independent experiments. Source data (**a**) are provided as a Source Data file.

mice (Shanghai Experimental Animal Centre of Chinese Academy of Sciences) were administered the fusion proteins at a dose of 5 mg/kg body weight by tail-vein injection. Mice were divided into 15 groups, corresponding to day 1–15. Blood was collected from the septum in heparin-containing tubes and then centrifuged to remove blood cells and to obtain plasma samples. The serum concentrations of the fusion proteins were determined by ELISA.

**Pseudovirus neutralization assay.** 293T cells and A549 cells were purchased from the American Type Culture Collection (ATCC, Manassas, VA). The identities of the cell lines were verified by STR analysis, and the cell lines were confirmed to be mycoplasma free. The cells were maintained in DMEM or 1640 with 10% fetal bovine serum. Cell culture media and supplements were obtained from Life Technologies, Inc. A well-established pseudovirus neutralization system[27–30] was adapted in our study. This assay is sensitive and quantitative, and can be conducted in biosafety level-2 facilities. Pseudoviruses containing the S glycoproteins from various viruses and a defective HIV-1 genome encoding luciferase as a reporter protein were prepared, and supernatants containing SARS-CoV or SARS-COV-2 pseudovirus were harvested 48 h post transfection and used for single-cycle infection of ACE2-transfected 293T and A549 cells (293T/ACE2 and A549/ACE2). The supernatants containing pseudovirus were preincubated with serially diluted indicated fusion proteins at 37 °C for 1 h before addition to cells. The culture was refed with fresh medium 24 h later and incubated for an additional 48 h. Luciferase activity was measured according to the manufacturer's instructions (Promega).

**Cell fusion inhibition assay.** A quantitative cell fusion assay based on β-galactosidase (β-gal) as a reporter gene was used to assess the neutralization activities of the fusion proteins[31]. 293T cells transfected with the indicated CoV S glycoprotein genes were preincubated with different fusion proteins at room temperature for 15 min, mixed with 293 T/ACE2 cells at a 1:1 ratio, and incubated at 37 °C for 4 h. The cells were then lysed, and the β-gal activity was measured. The protein concentrations during fusion were used to calculate the $IC_{50}$ value, which was defined as the concentration at which the β-gal activity was reduced by 50%.

**Reporting summary.** Further information on research design is available in the Nature Research Reporting Summary linked to this article.

## Data availability

The authors declare that the data supporting the findings of this study are available within the paper and its Supplementary Information files or from the corresponding author on reasonable request. The source data underlying Figs. 2a-b, 3a and Supplementary Figs. 1, 2 are provided as a Source Data file.

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

## Acknowledgements

We thank J Xu for critical support for the research. This study was supported by the National Natural Science Foundation of China (grant nos. 82041012, 81773261, 31970882, 81903140, and 81602690), the Shanghai Rising-Star Program (grant no. 19QA1411400), and the Shanghai Sailing Program (19YF1438600).

## Author contributions

C.L., K.Q., T.L., S.Z., and W.F. designed and performed the research; C.L., K.Q., T.L., S.Z., W.F., M.D., and S.H. analyzed the data; S.H. wrote the paper.

## Competing interests

M.D. is employed by Pharchoice Therapeutics Inc. (Shanghai) and is a shareholder at Pharchoice Therapeutics Inc. (Shanghai). The remaining authors declare no competing interests.
