## [Peer Review File · Nature Communications]

Reviewers' Comments:

Reviewer #1:

Remarks to the Author:

The novel coronavirus, SARS-CoV-2 (previously termed nCoV-2019), was recently shown to use the SARS-CoV receptor ACE2 for host cell entry. Wei and colleagues investigated whether recombinant ACE2 fused to IgG has antiviral activity. They show that ACE2-Ig binds to the S proteins of SARS-CoV-2 and SARS-CoV and inhibits cell-cell and virus-cell fusion driven by these glycoproteins. The results are clear but the concept of using recombinant ACE2 for treatment of coronavirus infection has previously been established and several points remain open.

Major

It is essential to extend the analysis of antiviral activity of ACE2-Ig to lung cell infection with authentic SARS-CoV-2. Moreover, it should ideally be determined whether passaging of SARS-CoV-2 in the presence of ACE2-Ig results in resistance development.

ACE2-Ig seems to inhibit SARS-CoV-2 S protein-driven entry with higher efficiency than SARS-CoV S protein-mediated entry. Why? In order to address this point, it is important to show on and off rates and KDs for the BiaCore analysis.

Minor

Data on the SARS-CoV-2 outbreak need to be updated, recent literature needs to be cited.

Reviewer #2:

Remarks to the Author:

The authors describe the design, generation and evaluation of a novel inhibitor of the SARS-CoV-2 infection based on soluble forms of its receptor ACE2. The idea to use soluble receptors for inhibition of viral infections is not new. It has been used for other viruses perhaps the most prominent is soluble CD4 for inhibition of HIV-1 infections.

However, the authors must be congratulated for executing a well thought plan so quickly ahead of other groups including ours. The also introduced two mutations to inhibit the enzymatic activity of AC2 in order to avoid or at least decreased possible side effects. The soluble ACE2 was fused to Fc to increase its half life in the circulation.

These constructs were tested in vitro using two assays – pseudovirus and cell fusion. In both assays the fusion proteins exhibited exceptionally potent activity with IC50s on the order 1 ug/ml. In mice those proteins also exhibited extended half life.

I didn't find any obvious problems with the design and execution of the experiments. The article is clearly written. There are only some minor typos and grammatical errors which could be corrected. An example is...M.D. declare that they are employees of Pharchoice Therapeutics, Inc. (Shanghai). M.D. is a shareholder at Pharchoice Therapeutics, Inc. (Shanghai)... In the first sentence the verb is in plural tense and in the second the same author is in singular tense although the subject is the same; suppose the second is correct.

In conclusion, this is an important timely study providing evidence for a high efficacy of a novel inhibitor of the SARS-CoV-2 which should be published as soon as possible.

Dimiter Dimitrov

Point-to-point responses to reviewers' comments

Reviewers' comments:

Reviewer #1 (Remarks to the Author):

The novel coronavirus, SARS-CoV-2 (previously termed nCoV-2019), was recently shown to use the SARS-CoV receptor ACE2 for host cell entry. Wei and colleagues investigated whether recombinant ACE2 fused to IgG has antiviral activity. They show that ACE2-Ig binds to the S proteins of SARS-CoV-2 and SARS-CoV and inhibits cell-cell and virus-cell fusion driven by these glycoproteins. The results are clear but the concept of using recombinant ACE2 for treatment of coronavirus infection has previously been established and several points remain open.

Major

It is essential to extend the analysis of antiviral activity of ACE2-Ig to lung cell infection with authentic SARS-CoV-2. Moreover, it should ideally be determined whether passaging of SARS-CoV-2 in the presence of ACE2-Ig results in resistance development.

Response: Very thanks to the suggestions. In the revised paper, we added lung cell line A549 in the pseudovirus neutralization assay. We know that the neutralization assay using live SARS-CoV-2 is supportive to our study, however, all the experiments involved the highly infectious SARS-CoV-2 have to be performed in biosafety level-3 facilities. Moreover, China's Ministry of Health recently required that all laboratories conducting or attempting to conduct live SARS-CoV-2 experiments undergo a detailed qualification and safety review for approval. According to the experimental facilities of the institution and the university, more hardware and administrative procedures may be required, and the time is difficult to estimate. Therefore, we regret to say that we are not able to complete such experiments immediately. Taking into account the current global spread of the SARS-CoV-2, we wish that our new results would be published as a peer reviewed paper as soon as possible, which will play a timely role in fighting the COVID-19. Moreover, the pseudovirus system has been widely adapted in lots of laboratories especially in SARS-related studies (Wong, S. K., et al. 2004; Yang, Z., et al. 2004; He, Y., et al. 2004; Zhang, H., et al. 2004, Zhu, Z., et al. 2007), this assay is sensitive and quantitative, and can be conducted in biosafety level-2 facilities. We also added a paragraph in the discussion section to describe the limitations of current paper.

ACE2-Ig seems to inhibit SARS-CoV-2 S protein-driven entry with higher efficiency than SARS-CoV S protein-mediated entry. Why? In order to address this point, it is important to show on and off rates and KDs for the BiaCore analysis.

Response: Very thanks to the suggestions. We quantify the kinetics of this interaction for ACE2-Ig with SARS-CoV-2 S protein and SARS-CoV S protein with surface plasmon resonance. Our data shows that SARS-CoV-2 S protein bound to the ACE2-Ig with ~10nM affinity, which is ~16-fold higher than SARS-CoV S protein binding to ACE2-Ig. The high affinity of ACE2-Ig binding to SARS-CoV2 may contribute to the apparent higher efficiency for ACE2-Ig to inhibit SARS-CoV-2

S protein-driven cell entry. We have added these results in the revised paper.

Reviewer #2 (Remarks to the Author):

The authors describe the design, generation and evaluation of a novel inhibitor of the SARS-CoV-2 infection based on soluble forms of its receptor ACE2. The idea to use soluble receptors for inhibition of viral infections is not new. It has been used for other viruses perhaps the most prominent is soluble CD4 for inhibition of HIV-1 infections.

However, the authors must be congratulated for executing a well thought plan so quickly ahead of other groups including ours. They also introduced two mutations to inhibit the enzymatic activity of ACE2 in order to avoid or at least decrease possible side effects. The soluble ACE2 was fused to Fc to increase its half life in the circulation.

These constructs were tested in vitro using two assays – pseudovirus and cell fusion. In both assays the fusion proteins exhibited exceptionally potent activity with IC50s on the order of 1 µg/ml. In mice those proteins also exhibited extended half life.

I didn't find any obvious problems with the design and execution of the experiments. The article is clearly written. There are only some minor typos and grammatical errors which could be corrected. An example is...M.D. declare that they are employees of Pharchoice Therapeutics, Inc. (Shanghai). M.D. is a shareholder at Pharchoice Therapeutics, Inc. (Shanghai)... In the first sentence the verb is in plural tense and in the second the same author is in singular tense although the subject is the same; suppose the second is correct.

In conclusion, this is an important timely study providing evidence for a high efficacy of a novel inhibitor of the SARS-CoV-2 which should be published as soon as possible.

Dimitar Dimitrov

Response: Very thank. The paper is revised for typos and grammatical errors.

Reviewers' Comments:

Reviewer #1:

Remarks to the Author:

The authors have partially addressed the questions of this reviewer. Literature citations are incomplete:

PMID: 32132184 showed that SARS-CoV-2 spike binds to ACE2 with 10-fold higher affinity than SARS-CoV spike and should be cited

PMID: 16007097 showed that soluble ACE2 can be used for SARS therapy and should be cited.

PMID: 32132184 showed that ACE2 is the receptor for SARS-CoV-2 and should be cited

Point-to-point responses to reviewers' comments

REVIEWERS' COMMENTS:

Reviewer #1 (Remarks to the Author):

The authors have partially addressed the questions of this reviewer. Literature citations are incomplete:

PMID: 32132184 showed that SARS-CoV-2 spike binds to ACE2 with 10-fold higher affinity than SARS-CoV spike and should be cited

PMID: 16007097 showed that soluble ACE2 can be used for SARS therapy and should be cited.

PMID: 32132184 showed that ACE2 is the receptor for SARS-CoV-2 and should be cited

Response: very thanks, we have added these references in the revised paper.